# A Layered and Parallelized Method of Eventual Model Checking

**Yati Phyo \*** , **Moe Nandi Aung** , **Canh Minh Do** **and Kazuhiro Ogata**

School of Information Science, Japan Advanced Institute of Science and Technology (JAIST),
Nomi 923-1292, Ishikawa, Japan; moenandiaung@jaist.ac.jp (M.N.A.); canhdo@jaist.ac.jp (C.M.D.);
ogata@jaist.ac.jp (K.O.)
**\*** Correspondence: yatiphyo@jaist.ac.jp

**Abstract:** Termination or halting is an important system requirement that many systems should satisfy and can be expressed in linear temporal logic as eventual properties. We devised a divide-and-conquer approach to eventual model checking in order to reduce the state space explosion in model checking. The idea of the technique is to split an original model checking problem for eventual properties into multiple smaller model checking problems and handle each smaller one. Due to the nature of the divide-and-conquer approach, each smaller model checking problem can essentially be tackled independently. Hence, this paper proposes a parallel technique/tool based on a master–worker pattern for the divide-and-conquer approach to model checking eventual properties. We carry out some experiments to show the effectiveness of our parallel technique/tool, which can somewhat enhance the running performance to a certain extent when conducting model checking for eventual properties.

**Keywords:** eventual properties; model checking; master–worker pattern; parallelization; Maude





## 1. Introduction

In addition to the traditional testing techniques that have been in use for decades, model checking [1] is one promising approach for increasing the reliability of software and hardware systems. By checking the entire reachable state spaces of the systems, model checking can systematically verify that finite-state systems satisfy some desired properties. The state space explosion, on the other hand, is well-known to be the most difficult problem in model checking due to the fact that the number of states in a system under the verification's reachable state space often increases exponentially. Although numerous techniques have been suggested to improve the situation, the issue still needs to be resolved. These include abstraction [2–4] and partial order reduction [5]. Aside from that, there is a running performance issue for model checking because a lot of time is often spent performing model checking experiments when the reachable state space is huge. One of the main approaches to cope with the running performance issue is to parallelize model checking to take advantage of multicore architectures.

Our research team devised a divide-and-conquer approach to model checking some specific properties such as conditional stable properties [6], leads-to properties [7], eventual properties [8], and until stable properties [9] in order to cope with the state space explosion problem in model checking. We can formalize those properties in linear temporal logic (LTL); however, we need to handle each property separately because it is challenging to find a single technique in order to cope with four different properties at once to reduce the problem. The core principle of our approach is to split the original model checking problem for each property into multiple smaller model checking problems and work on each smaller one. Suppose that the sub-state space's size for each smaller problem is substantially smaller compared to the reachable state space's size for the original model checking problem. In this situation, our approach can somewhat reduce the state space explosion problem for

each property to a certain extent. Then, for each of the conditional stable properties [10], leads-to properties [11], and eventual properties [12], our research team constructed a sequential tool. Due to the nature of the divide-and-conquer approach, each smaller model checking problem can essentially be tackled independently. Therefore, our research team has proposed parallel techniques/tools for conditional stable properties [10] and leads-to properties [13]. In this present paper, we concentrate on parallelizing model checking for eventual properties to benefit from multicore architecture in order to enhance the running performance for model checking eventual properties, in addition to easing the state space explosion as demonstrated in [12].

Informally, eventual properties indicate that something will eventually occur. This paper only copes with the eventual properties expressed in LTL as $\Diamond \varphi$, where $\varphi$ is restricted to a state proposition. The properties can be used to define a number of significant software needs, including termination or halting, which is an essential software requirement that numerous systems should meet. We devised a divide-and-conquer approach to eventual model checking (referred to as DCA2EMC from now on) [8] that intends to reduce the state space explosion and constructed a sequential tool [12] in support of the approach. Some experiments have been carried out that show that our approach is a promising solution to reduce the state space explosion problem for eventual model checking. Due to the nature of the divide-and-conquer approach of DCA2EMC, we would like to leverage parallelization to address the ongoing performance issue in model checking for eventual properties. This paper is an extended version of our conference paper [12], where we propose a technique to parallelize model checking for eventual properties and construct a support tool. The effectiveness of our proposed technique/tool is demonstrated by carrying out some experiments.

The sequential tool [12] has been developed in Maude [14], as has the parallel tool. Maude is a high-performance specification/programming language relying on rewriting logic [15]. Many parallel applications have been developed using Maude, including parallel tools for model checking conditional stable properties [10] and leads-to properties [13] based on a divide-and-conquer approach, and the parallelization of Maude-NPA for analyzing cryptographic protocols [16]. SPIN [17] is a well-known model checker widely used in academia and industry. From the perspective of running performance and memory usage, Maude and SPIN are comparable [18]. Therefore, it is worth considering Maude in this work. Moreover, Maude supports reflective programming (or meta-programming), making it easier to extend Maude facilities and use Maude as a handy software component in our implementation. Therefore, we employ Maude to develop the parallel tool rather than another programming language. The case studies and the parallel tool utilized in our experiments are both found at https://github.com/yatiphyo/DCA2MC (accessed on 29 May 2023).

The remainder of the present paper is organized as follows. Section 2 mentions some preliminaries. Section 3 describes the algorithm for DCA2EMC. Section 4 describes the parallelization of DCA2EMC. Section 5 reports our experimental results. Section 6 mentions some existing work. Finally, Section 7 concludes the paper with some avenues for future work.

## 2. Preliminaries

A Kripke structure $K$ is a tuple $< S, I, T, A, L >$ of five elements as follows: (1) a set of states denoted by $S$, (2) a set of initial states denoted by $I$ such that $I \in S$, (3) a left-total binary relation over states $S$ denoted by $T$, (4) a set of atomic propositions denoted by $A$, and (5) a labeling function from $S$ to $2^A$ denoted by $L$. $L(s)$ denotes a set of atomic propositions that are true in $s$ for each $s \in S$. Each $(s, s') \in T$ may be thought of as a state transition, which may be expressed as either $s \rightarrow s'$ or $s \rightarrow_K s'$ if $K$ is clear from the context. $s_0, \ldots, s_i, s_{i+1}, \ldots$ is called an infinite sequence of states or a path (referred to as $\pi$) if $(s_i, s_{i+1})$ is a state transition for each $i = 0, \ldots, i, \ldots$. The following notations for paths are used:

- $\pi^i \triangleq s_i, s_{i+1}, \ldots,$
- $\pi_i \triangleq s_0, \ldots, s_{i-1}, s_i, s_i, \ldots,$
- $\pi(i) \triangleq s_i,$

where $\pi^i$ is a postfix of $\pi$ from the $i$th state in $\pi$; $\pi_i$ is a prefix of $\pi$, where the $i$th state $s_i$ of $\pi$ is repeated forever in the prefix; and $\pi(i)$ is the $i$th state in $\pi$, where $s_0$ denotes the first state (or the 0th state) in $\pi$. If $\pi(0) \in I$, we refer to a path $\pi$ as a computation. $\mathcal{P}$ and $\mathcal{C}$ are used to represent sets of all paths and computations, respectively. By definition, we have $\mathcal{C} \subseteq \mathcal{P}$. The collection of distinct paths that begin with $s \in S$ is denoted by $\mathcal{P}_{(K,s)}$. Given a natural number $b$, $\mathcal{P}^b_{(K,s)}$ is the collection of paths $\pi_b$ such that $\pi \in \mathcal{P}_{(K,s)}$. We employ $\mathcal{P}^\infty_{(K,s)}$ to indicate $\mathcal{P}_{(K,s)}$.

Assuming that $p$ is an atomic proposition, we define the syntax of an LTL formula $\varphi$ in the following manner:

$$\varphi ::= \top \mid p \mid \neg\varphi \mid \varphi \vee \varphi \mid \bigcirc \varphi \mid \varphi \, \mathcal{U} \, \varphi$$

The set of all LTL formulas is denoted by $\mathcal{F}$. Inductively, we can define $K, \pi \models \varphi$ as follows for $\pi \in \mathcal{P}$ and $\varphi \in \mathcal{F}$:

- $K, \pi \models \top$;
- $K, \pi \models p$ iff $p \in L(\pi(0))$;
- $K, \pi \models \neg\varphi_1$ iff $K, \pi \not\models \varphi_1$;
- $K, \pi \models \varphi_1 \vee \varphi_2$ iff $K, \pi \models \varphi_1$ and/or $K, \pi \models \varphi_2$;
- $K, \pi \models \bigcirc \varphi_1$ if and only if $K, \pi^1 \models \varphi_1$;
- $K, \pi \models \varphi_1 \, \mathcal{U} \, \varphi_2$ if and only if there exists a natural number $i$ such that $K, \pi^i \models \varphi_2$ and for all natural numbers $j < i$, $K, \pi^j \models \varphi_1$.

Here, the LTL formulas $\varphi_1$ and $\varphi_2$ are used. Then, we have $K \models \varphi$ iff $K, \pi \models \varphi$ for all computations $\pi \in \mathcal{C}$ of $K$. The next temporal connective and the until temporal connective are defined as $\bigcirc$ and $\mathcal{U}$, respectively.

We use some other logical and temporal connectives defined as follows:

- $\bot \triangleq \neg\top$;
- $\varphi_1 \wedge \varphi_2 \triangleq \neg(\neg\varphi_1 \vee \neg\varphi_2)$;
- $\varphi_1 \Rightarrow \varphi_2 \triangleq \neg\varphi_1 \vee \varphi_2$;
- $\Diamond \varphi \triangleq \top \, \mathcal{U} \, \varphi$;
- $\Box \varphi \triangleq \neg(\Diamond \neg\varphi)$.

$\Box$ is the always temporal connective, while $\Diamond$ is the eventual temporal connective. State propositions do not include any temporal connectives as propositional formulas in propositional logic, and then the first state $\pi(0)$ can determine whether a path $\pi$ satisfies a state proposition. In this present paper, the properties in the form of $\Diamond\varphi$ are called eventual properties, where $\varphi$ is restricted to a state proposition.

In this present paper, if a collection's non-empty constructor is both associative and commutative (AC), it is referred to as a soup. An observable component is a pair of names and values (e.g., $(\text{loc}[p] : \text{ws})$ indicating that process $p$ is resided at ws or the waiting section). States are characterized by using some observable components. Formally, a state is specified as a braced soup of observable components. For example, if three observable components $oc_1, oc_2, oc_3$ are used to characterize a state for a particular purpose, $\{oc_1 \, oc_2 \, oc_3\}$ is the formalization of a state, where the juxtaposition operator (or the empty syntax) is utilized as the non-empty constructor of soups. Because the order does not matter in AC collections, $\{oc_1 \, oc_2 \, oc_3\}$ is equivalent to other permutations of $oc_1, oc_2, oc_3$, such as $\{oc_2 \, oc_3 \, oc_1\}$. Rewrite rules can be used to describe state transitions. In particular, we formalize systems and protocols as Kripke structures in Maude [14] and then we use a built-in LTL model checker in Maude for conducting model checking experiments.

### 3. A Divide-and-Conquer Approach to Eventual Model Checking

Given a Kripke structure $K$ and an eventual property, DCA2EMC divides the original reachable state space that starts from each $s_0 \in I$ into $L + 1$ layers as shown in Figure 1 for model checking the eventual property in a layered way. We call intermediate layers those from 0 to $L$, while we call the final layer $L + 1$. Let $d_l$ be a positive natural number to indicate the depth of each layer $l$ for $l = 1, \ldots L$, $d_0$ be 0, and $d_{L+1}$ be $\infty$. A sequence $d_1 \, d_2 \, \ldots \, d_L$ is referred to as a layer configuration used for DCA2EMC. Let $d(l)$ be $d_0 + d_1 + \ldots + d_l$ indicating the true depth of layer $l$ from each initial state in $I$ for $l = 1, \ldots, L$, $d(0)$ be 0, and $d(L + 1)$ be $\infty$. States residing at depth $d(l)$ or the bottom of layer $l$ are utilized as the beginning states (or initial states) of layer $l + 1$ in DCA2EMC. If there are $n_l$ states residing at the bottom of layer $l$, then layer $l + 1$ has $n_l$ sub-state spaces for each $l = 0, 1, \ldots, L$. Note that for each state, its sub-state space is the reachable state space that starts from the state with a bounded depth, and an unbounded depth is used for the final layer. In this manner, we split the reachable state space from $s_0$ into multiple smaller sub-state spaces. We carry out model checking experiments with a Kripke structure $K_l$ for sub-state spaces residing in each intermediate layer $l$. $K_l$ is constructed from $K$ by removing all state transitions that come out from the states residing at the bottom of layer $l$ and adding a self-transition to each state. Because the final layer's depth is $\infty$, we simply carry out model checking experiments with $K$ for sub-state spaces residing in the final layer.

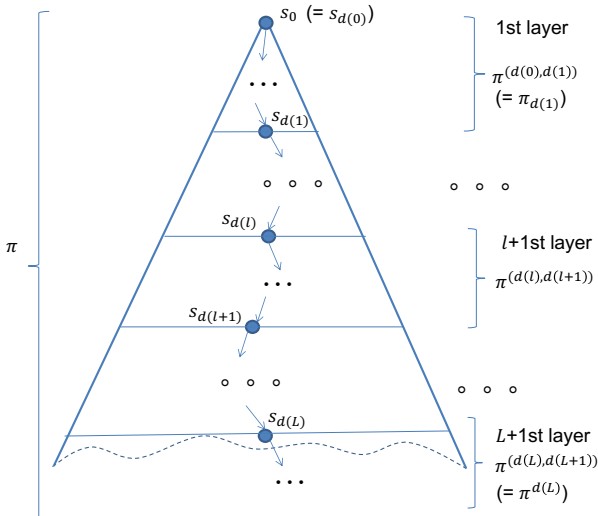

**Figure 1.** Division of the original reachable state space into $L + 1$ layers.

We can check $K \models \Diamond \varphi$, where $\varphi$ is restricted to state propositions in a layered way, as presented in Algorithm 1. For each intermediate layer $l \in \{0, 1, \ldots, L\}$, DCA2EMC needs to collect all counterexample (or $\mathtt{cx}$ for short) states that do not satisfy $\Diamond \varphi$. Let $ES$ and $ES'$ be two sets of states used to collect all $\mathtt{cx}$ states at each intermediate layer. Our procedure is as follows:

- $l = 0$, $ES$ is initially $I$ in the code snippet at line 1, consisting of all initial states, and is regarded as $\mathtt{cx}$ states in our initialization.
- For each $l \in \{1, \ldots, L\}$, we first check if $ES$ is empty in the code snippet at lines 3–5. If so, Success is returned; otherwise, we keep going as follows. $ES'$ is then set to an empty set in the code snippet at line 6. Model checking an eventual property for the sub-state space that starts from each state in $ES$ with $K_l$ is conducted. If a $\mathtt{cx}$ is found, we add the last state of the $\mathtt{cx}$, which must have a self-transition because of $K_l$, to $ES'$ in the code snippet at lines 7–13. In the end, all $\mathtt{cx}$ states for layer $l$ are stored in $ES'$ and assigned to $ES$ in the code snippet at line 14.

Note that all states in $ES$ or $ES'$ are called $\mathtt{cx}$ states in this paper.

For the final layer $L + 1$, DCA2EMC needs to be carried out as follows:

- Model checking an eventual property for the sub-state space that starts from each state in *ES* with *K* is conducted. If a `cx` is found, it returns Failure in the code snippet at lines 16–22. In other words, a `cx` of the original model checking problem is found.
- If no `cx` was found in the end, Success is returned in the code snippet at line 23.

To guarantee the correctness of the algorithm for DCA2EMC, we have proved a theorem in [8]. As shown in Algorithm 1, it just returns `Failure` when $K \not\models \Diamond\varphi$, but with a small modification as described in [8], we can construct and return a `cx`. Both the sequential version of DCA2EMC in [12] and the parallelization of DCA2EMC in this present paper are capable of showing a `cx` when $K \not\models \Diamond\varphi$. For a comprehensive understanding of how DCA2EMC operates using a simple example, readers can check Sect. IV in [8] for more detailed information.

---

**Algorithm 1:** A divide-and-conquer approach to eventual model checking in a layered way.

---

**input** : $K$ – a Kripke structure
$\varphi$ – a state proposition
$L$ – a positive natural number
$d_1\ d_2\ \dots\ d_L$ – a layer configuration
**output:** Success ($K \models \Diamond\varphi$) or Failure ($K \not\models \Diamond\varphi$)

1   $ES \leftarrow I$
2   **forall** $l \in \{1, \dots, L\}$ **do**
3     **if** $ES = \varnothing$ **then**
4       **return** Success
5     **end**
6     $ES' \leftarrow \varnothing$
7     **forall** $s \in ES$ **do**
8       **forall** $\pi \in \mathcal{P}^{d_l}_{(K_l, s)}$ **do**
9         **if** $K_l, \pi \not\models \Diamond\varphi$ **then**
10          $ES' \leftarrow ES' \cup \{\pi(d_l)\}$
11         **end**
12       **end**
13     **end**
14     $ES \leftarrow ES'$
15   **end**
16   **forall** $s \in ES$ **do**
17     **forall** $\pi \in \mathcal{P}_{(K, s)}$ **do**
18       **if** $K, \pi \not\models \Diamond\varphi$ **then**
19         **return** Failure
20       **end**
21     **end**
22   **end**
23   **return** Success

---

## 4. A Parallel Version of DCA2EMC

As shown in Algorithm 1, finding counterexample states in each sub-state space at each intermediate layer of DCA2EMC can essentially be conducted independently; however, the overhead to parallelize this part may be higher than the benefit that can be obtained from parallelization, as demonstrated in [19]. Meanwhile, the model checking problem for each sub-state space residing in the final layer of DCA2EMC can be essentially conducted independently. Therefore, this section describes how to build a parallelization of DCA2EMC relying on a master–worker pattern, where we only conduct the model

checking experiments at the final layer in parallel, while collecting all counterexample states residing at the bottom of each intermediate layer up to layer $L$, which proceeds in a serial way. Object-oriented systems, in which objects communicate with one another through Maude Sockets, are supported by Maude. Hence, we use these facilities in Maude to develop the parallel tool.

The parallel tool is constructed using a master–worker pattern, where one master and many workers participate. The master is responsible for constructing jobs, distributing jobs to workers, and terminating the parallel tool whenever either a `cx` is found by a worker at the final layer or workers have performed all jobs. The master and the workers communicate using three different types of messages: job, getJob, and stop. A job message is constructed from a state $s_{d(L)}$ residing at the bottom of layer $L$ and a log list of the state, which is a list $\langle s_{d(L-1)} : d(L) \rangle \ldots \langle s_{d(1)} : d(2) \rangle \langle s_{d(0)} : d(1) \rangle$ of pairs of states and natural numbers. The log list is used to track the path that starts from the initial state to the current state to build the `cx` of the original model checking problem if applicable. Only the master is the person sending a job message to a worker. getJob and stop messages are literal strings sent from workers to the master. Each worker is responsible for carrying out a model checking experiment with the eventual property concerned from the state encapsulated in each job that the master assigns to the worker. If no `cx` is found, the worker asks the master for a new job by sending a getJob message. Otherwise, the worker finds a `cx` and then it sends a stop message for terminating the parallel tool and constructs a full `cx` for the original model checking problem based on the log list encapsulated in the job and the `cx` found in the model checking experiment at the final layer. The reader can refer to [12] on how to construct the full `cx` for more details.

Algorithm 2 presents the pseudocode for assigning jobs carried out by the master. The master maintains two queues: one for jobs (referred to as *jobs*) and the other for worker identifiers (referred to as *workers*) to allocate jobs to workers in an equitable manner. Initially, the master is responsible for generating all `cx` states residing at the bottom of each intermediate layer up to layer $L$ using the *findCxStates* function in the code snippet at line 1 with the initial state $s_0$ and the layer configuration $d_1 \ldots d_L$ as its parameters. Because this step is conducted in a serial way, we build the function based on how we generate `cx` states for each intermediate layer implemented in the sequential tool [12]. Recall that each job in *jobs* is constructed from a state residing at the bottom of layer $L$ and a log list of the state. The master then carries out the above-described duties. In the code snippet at lines 3–11, the master checks if there is any message from the workers. If that is the case, the master checks whether the message is either a getJob message or a stop message. If it is the former, the worker identifier is added to the *workers* queue; otherwise, the master disconnects all workers and returns Failure. In the code snippet at lines 12–16, the master checks if neither *workers* nor *jobs* is empty. If so, the master delivers a job to a worker. In the code snippet at lines 17–20, the master verifies whether workers have completed all jobs in *jobs*. If so, the master disconnects all workers and returns Success.

Algorithm 3 shows the pseudocode for handling jobs operated by each worker. On the first line of the code snippet, the workers send a very first getJob message to the master to ask for a job. The code snippet at lines 2–11 shows what the worker is supposed to do as described above. When the worker is open, it needs to do as follows. The worker receives a job message in the form of a state and its log list from the master in the code snippet at line 3. The worker then carries out its above-described duties in the code snippet at lines 4–10. If a `cx` is found, a full `cx` is constructed and returned; otherwise, the worker asks the master for a new job by sending a getJob message. The worker is not open only when the master disconnects all workers. If so, the worker is subsequently terminated.

---

**Algorithm 2:** The master delivers jobs to workers

---

   **input** : $K$ – a Kripke structure

           $s_0 \in I$ – an initial state of $K$

           $\varphi$ – a state proposition

           $d_1 \dots d_L$ – a layer configuration

           $N$ – #workers

   **output**: Success ($K$, $s_0 \not\models \Diamond\varphi$) or Failure ($K$, $s_0 \not\models \Diamond\varphi$)

**1** $jobs \leftarrow findCxStates(s_0, d_1 \dots d_L)$;

**2** **while** *True* **do**

**3**    **for** $k \leftarrow 1$ **to** $N$ **do**

**4**       **if** $MSG \leftarrow rec(worker_k)$ **then**

**5**          **if** $MSG = stop$ **then**

**6**            closeConnect();

**7**            **return** Failure;

**8**          **else if** $MSG = getJob$ **then**

**9**            $enq(workers, worker_k)$;

**10**       **end**

**11**    **end**

**12**    **while** $\neg$ *isEmpty(workers)* $\wedge \neg$ *isEmpty(jobs)* **do**

**13**       $worker \leftarrow deq(workers)$;

**14**       $job \leftarrow deq(jobs)$;

**15**       snd(worker, job);

**16**    **end**

**17**    **if** *isEmpty(jobs)* $\wedge$ *size(workers)* $= N$ **then**

**18**       closeConnect();

**19**       **return** Success;

**20**    **end**

**21** **end**

---

**Algorithm 3:** Workers process jobs

---

   **input** : $K$ – a Kripke structure

           $\varphi$ – a state proposition

           $d_1 \dots d_L$ – a layer configuration

   **output**: a counterexample if there is any

**1** send(master, getJob);

**2** **while** *isOpen()* **do**

**3**    **if** $(s_{d(L)}, log) \leftarrow rec(server)$ **then**

**4**       **forall** $\pi \in \mathcal{P}_{(K, s_{d(L)})}$ **do**

**5**          **if** $K, \pi \not\models \Diamond\varphi$ **then**

**6**            send(master, stop);

**7**            **return** *constructCx()*;

**8**          **end**

**9**       **end**

**10**       send(master, getJob);

**11**    **end**

**12** **end**

---

## 5. Experiments

We employed a MacPro that carries a 2.5 GHz microprocessor with 28 cores and 768 GB memory of RAM to carry out some experiments for the sequential version of DCA2EMC (referred to as the sequential tool), the parallelization of DCA2EMC (referred

to as the parallel tool), and the Maude LTL model checker. The Maude LTL model checker can be compared with SPIN from the perspective of running performance and memory usage, as demonstrated in [20]. Hence, it is worth comparing the running performance of the sequential and parallel tools with that of the Maude LTL model checker. We employ four mutual exclusion protocols and one autonomous vehicle intersection control protocol as case studies: Qlock, Anderson, MCS, TAS, and LJPL [21]. For the mutual exclusion protocol known as Anderson [22], a shared atomic array is used by all processes. For the list-based queuing mutual exclusion MCS [23], created by Mellor-Crummey and Scott, it has been adapted for usage with Java virtual machines. Note that TAS is the simplest mutual exclusion protocol in these experiments that employs an atomic operator test&set. For four mutual exclusion protocols, we suppose that processes enter the critical section no more than once and the same property $\lozenge$ `inFs1` is checked for four mutual exclusion protocols. For the LJPL protocol [21], we would like to check if all vehicles will eventually cross the intersection.

The pseudocode of each of Anderson, Qlock, MCS, and TAS and its description can be found in [8] for more details. For the LJPL protocol [21], it enables us to control all vehicles passing through the intersection without colliding, which is shown in Figure 2. The right-hand side of a street is where vehicles are supposed to run and there are two lanes on each side. According to Figure 2, each lane is given a name. Vehicles on the right lane of one side of a street, such as `lane0`, are supposed to go straight or turn right when crossing the intersection, as depicted in Figure 2. Meanwhile, those on the left lane of one side of a street, such as `lane1`, are supposed to turn left, as depicted in Figure 2. The area where the two streets overlap is the critical section, where vehicles should be controlled to prevent collisions. Vehicles on `lane0` and `lane4` are permitted to pass through the intersection simultaneously, while those on `lane0` and `lane2` are not. `lane0` and `lane4` are concurrent, while `lane0` and `lane2` are in conflict, and similarly for other lanes. The reader is referred to [21] in order to understand in detail how the protocol controls vehicles passing through the critical section without colliding.

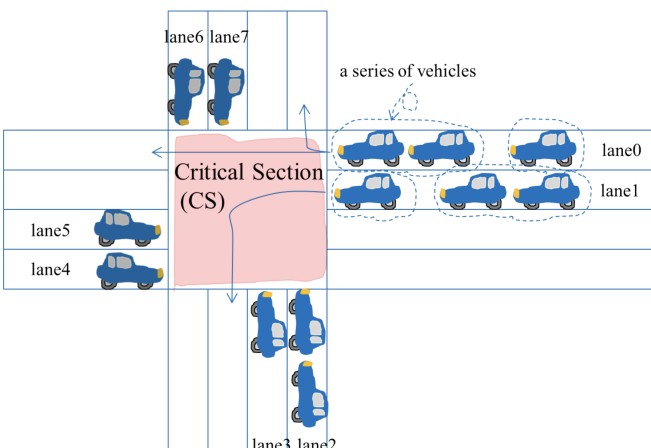

**Figure 2.** The intersection in the LJPL autonomous vehicle intersection control protocol.

*5.1. Comparison of the Sequential and Parallel Tools, and Maude LTL Model Checker*

For four protocols, we carried out some model checking experiments for eventual ($\lozenge$ `inFs1`) properties for Qlock with 10 processes, Anderson with 9 processes, MCS with 6 processes, and TAS with 13 processes. For the LJPL protocol, we carried out some model checking experiments to check if six vehicles eventually pass through the intersection with the following configuration: no vehicle is on `lane1`, `lane5`, and `lane7`, one vehicle is on each lane of `lane0`, `lane2`, `lane3`, and `lane4`, and two vehicles are on `lane6`. Table 1 shows our experimental data. The Layers column with $d_1\ d_2\ \ldots\ d_L$ information indicates that DCA2EMC is employed and $d_i$ is the depth of the $i$th layer. In these experiments,

we used two optimization techniques proposed in [19] with the sequential and parallel tools in order to find a good layer configuration and enhance the running performance on generating counterexample states up to the final layer for each case study. The sequential tool presents an impressively better running performance than the Maude LTL model checker for Qlock, Anderson, MCS, and LJPL. The parallel tool can speed up by up to 9.1 times, 6.1 times, 5.0 times, 2.3 times, and 5.5 times faster than the sequential tool for Qlock, Anderson, TAS, MCS, and LJPL, respectively, when eight workers are used. The improvement from parallelization is understandable because we need to pay additional costs, such as those associated with master–worker communication. Moreover, we only conduct parallelization for the sub-state spaces residing in the final layer and so the improvement in MCS is not impressive compared with other case studies because it needs to spend a lot of time generating counterexample states up to the final layer. For TAS, the Maude LTL model checker presents a better running performance than both the sequential tool and the parallel tool when we use eight workers for the parallel tool. In comparison to the other three mutual exclusion protocols used in these experiments, TAS is the most simple. The key distinction between TAS and the others is that in TAS, all processes awaiting entry into the critical section have equal opportunity to do so, but in the others, only one such process can do so. This creates a symmetry of processes in TAS and so many sub-state spaces may share many of the same states in the final layer. Consequently, the sequential and parallel tools need to visit such states repeatedly, degrading the running performance. This is why the sequential tool and the parallel tool of DCA2EMC cannot surpass the Maude LTL model checker, although the parallel tool can run up to 5.0 times faster than the sequential tool for the TAS case study. Using the symmetric reduction technique [24] in conjunction with our techniques would be one potential solution to address the problem, which we leave for future work. In summary, our parallel tool can greatly enhance the running performance compared with the sequential tool for five protocols, and significantly surpass the Maude LTL model checker for four protocols, but not for the simplest mutual exclusion protocol TAS.

**Table 1.** Running performance of the sequential and parallel tools, and the Maude LTL model checker for eventual model checking.

| Protocol | Sequential Tool (DCA2EMC) | Parallel DCA2EMC (8 Workers) | Layers | Maude LTL Model Checker |
|---|---|---|---|---|
| Qlock (10 processes) | 3 h 12 m 12 s | 21 m 1 s | 2 2 | 10 d 11 h 31 m 8 s |
| Anderson (9 processes) | 27 m 19 s | 4 m 29 s | 2 2 | 1 d 3 h 18 m 43 s |
| MCS (6 processes) | 1 d 15 h 32 m 21 s | 17 h 10 m 14 s | 4 4 4 4 2 | 5 d 20 h 52 m 47 s |
| TAS (13 processes) | 2 d 4 h 19 m | 10 h 26 m 17 s | 3 3 3 | 2 h 28 m 29 s |
| LJPL (6 vehicles) | 59 m 20 s | 10 m 48 s | 2 2 2 | 8 h 3 m 58 s |

### 5.2. Scalability of the Parallel Version of DCA2EMC

We would like to measure the scalability of the parallelization of DCA2EMC with the use of different numbers of workers in these experiments. We only select the TAS case study with 13 processes to conduct model checking experiments with the parallel tool when 8, 16, 24, and 32 workers are used. This is because when we use eight workers for the TAS case study, the parallel tool cannot surpass the Maude LTL model checker. Let us recall that the verification time for DCA2EMC and the Maude LTL model checker in these experiments is 2 d 4 h 19 m and 2 h 28 m 29 s, respectively, and the layer configuration used is 3 3 3 in these experiments. We would like to see how much improvement we can

achieve from the parallel tool when we increase the number of workers for the TAS case study. Table 2 shows the experimental results, which are also represented in the graph in Figure 3. We only employ up to 32 workers because the MacPro only has 28 cores. The dashed line represents the perfect speedup, whereas the line with diamond symbols is the actual speedup. The ratio of the times spent using the sequential and parallel tools is used to compute the speedup. Finding a sweet point from which the verification time later degrades is the goal of scaling measurement. In these experiments, we were unable to conduct experiments with many workers due to the limitation of the MacPro. However, the increase in speed that results from adding more workers has a tendency to rise more or less steadily and approach the peak at the end. Then, if we continue adding workers, the increase in speed will diminish because there is a high communication overhead and a decreasing workload for each worker. In these experiments, the parallel tool can speed up by up to 5.0 times, 8.8 times, 11.9 times, and 13.5 times faster than the sequential tool when the numbers of workers are 8, 16, 24, and 32, respectively, as plotted in Figure 3. In DCA2EMC, model checking sub-state spaces residing in the final layer are independent and there are many counterexample states. Thus, we anticipate that many workers may be needed to reach the peak for TAS. At that point, the parallel tool may surpass the Maude LTL model checker even for TAS case study. Our future work includes using a supercomputer to conduct model checking experiments with the parallel tool that can use as many workers as possible.

**Table 2.** Scalability of the parallelization of DCA2EMC for TAS.

| Protocol | #Workers | Running Performance |
|---|---|---|
| TAS (13 processes) | 8 | 10 h 26 m 17 s |
| | 16 | 5 h 55 m 21 s |
| | 24 | 4 h 23 m 20 s |
| | 32 | 3 h 52 m 49 s |

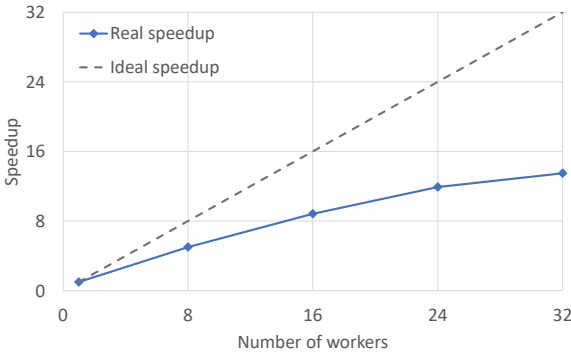

**Figure 3.** Scalability speedup for TAS.

## 6. Related Work

SAT/SMT-based bounded model checking (or SAT/SMT-BMC for short) [25] is an extremely successful method for dealing with the state space explosion in model checking. It can find a counterexample at the position close to an initial state, but proving that a system has the desired properties is impossible. To address this limitation, *k*-induction [26,27], an extension of SAT/SMT-BMC, has been proposed in which the technique uses a combination of mathematical induction and SAT/SMT-BMC in order to show the satisfaction of a system with its required properties. An SAT/SMT solver is used to verify the desired properties up to a certain depth (or a bounded depth) from an arbitrary initial state, and this step is regarded as the base case. For each state sequence up to the bounded depth, all successor states from the last state of the state sequence are verified with the desired properties using an SAT/SMT solver, and this step is regarded as the induction step. Our technique [6]

shares the basic idea with SAT/SMT-BMC when we conduct bounded model checking experiments for sub-state spaces residing at each intermediate layer. DCA2EMC can also be regarded as an extension of BMC because we conduct unbounded model checking for sub-state spaces residing in the final layer in order to prove the desired properties, but no SAT/SMT solvers are used.

Barnat et al. [28] have surveyed recent developments in parallel algorithms for LTL model checking. To optimize the use of multi-core architectures, we are required to modify graph search algorithms. DiVinE 3.0 [29] and a multi-core extension of SPIN [30] are two parallel model checkers that use these algorithms. In contrast, we can employ any existing LTL model checker to implement a parallelization of DCA2EMC, without the need to modify graph search algorithms. This is a key distinction from any parallel LTL model checker currently available.

Inverso et al. [31] expanded the use of SAT/SMT-based BMC to verify concurrent programs. They used an unwinding (or unfolding) bound, denoted by $u$, and a round-robin schedule, denoted by $r$. The first step is to transform a concurrent program $P$ into an intermediate program $P_u$, where they unfold all loops and inline function calls with $u$ as a bound, while those used for creating threads are disregarded. The second step is to convert $P_u$ into a sequential program $Q_{u,r}$ by enumerating all behaviors of $P_u$ in $r$ round-robin schedules. The third step is to encode $Q_{u,r}$ into a propositional formula, which can be checked with a SAT/SMT solver. To parallelize the analysis of such a propositional formula, it can be split into multiple sub-formulas, assigned to multiple instances running a SAT/SMT solver, and handled in parallel [32]. It seems that their approach can only cope with safety properties, while our approach can cope with eventual properties, which is a category of liveness properties.

A distributed-memory version of SPIN has been presented by Lerda and Sisto [33]. Their proposed approach is similar to ours as it addresses the problem of formal specifications for systems under verification becoming too large to fit into a computer's physical memory. This leads to a significant decrease in performance and in the worst case it is even impossible to conduct model checking experiments. To solve this issue, for a large-state system, they split the whole reachable state space into smaller sub-state spaces and assign them to multiple nodes or computers connected through the network. The advantage of their approach is the ability to integrate some existing optimization techniques from SPIN, such as bit state hashing and partial order reduction, into their approach. Some case studies have been conducted, showing the efficiency of their proposed approach. However, the distributed-memory version of SPIN can only cope with safety properties, while our parallelization of DCA2EMC can cope with eventual properties, which is a category of liveness properties.

When verifying huge systems, using an exhaustive search with SPIN is impossible. It then provides a bit-state verification mode, which conducts a non-exhaustive search on the whole reachable state space to tackle such huge systems. However, the likelihood that the SPIN bit-state verification mode will miss system defects increases with the size of the system under verification. To address this limitation, Swarm Verification [34] has been introduced. In this technique, parallelism and the diversity of search strategies are two main principles. One instance of bit-state verification is carried out for each search strategy. These instances can be run simultaneously and are essentially independent. By using various search strategies, we may explore different regions of the reachable state space, increasing the likelihood of flaw detection, as well as improving coverage. Grapple [35], a GPU-based Swarm Verification implementation, has also been proposed since then. In our approach, the reachable state space from each initial state is split into multiple layers, generating multiple sub-state spaces, and each sub-state space is checked exhaustively with the Maude LTL model checker. It would be beneficial to employ the Swarm Verification idea in conjunction with our technique so that it can use the Swarm Verification for each sub-state space rather than an exhaustive search to potentially uncover flaws in huge systems more quickly.

## 7. Conclusions

We have described the parallelization of DCA2EMC with a master–worker pattern. The parallel tool is implemented in Maude, where one master and several workers participate and communicate with each other through Maude sockets. For DCA2EMC, only model checking sub-state spaces residing in the final layer are conducted in parallel, while generating all counterexample states at each intermediate layer up to layer $L$ is conducted in a serial way. Some case studies have been carried out to demonstrate the effectiveness of our parallel technique/tool, which can reduce the running performance issue to some extent when conducting model checking for eventual properties.

In order to utilize our proposed techniques and tools effectively for eventual model checking, it is crucial to create a formal specification of the system under verification such that each sub-state space has significantly fewer states than the overall states of the original reachable state space of the system. This requires eliminating long lasso loops (or long backward loops), which could avoid situations where some sub-state spaces residing in the final layer have significantly fewer states than the overall states of the original reachable state space. Dealing with long lasso loops efficiently is a challenge for our approach, and developing a technique to handle them effectively is one avenue for our future work. Moreover, we would like to conduct more case studies to demonstrate the effectiveness of our techniques/tools.

**Author Contributions:** Conceptualization, K.O.; methodology, Y.P. and C.M.D.; software, Y.P. and C.M.D.; validation, Y.P. and M.N.A.; formal analysis, Y.P. and M.N.A.; investigation, Y.P. and M.N.A.; resources, Y.P. and M.N.A.; data curation, Y.P. and C.M.D.; writing—original draft preparation, Y.P.; writing—review and editing, Y.P., M.N.A., C.M.D. and K.O.; visualization, Y.P. and M.N.A.; supervision, K.O.; project administration, K.O.; funding acquisition, K.O. All authors have read and agreed to the published version of the manuscript.

**Funding:** This work was partially supported by JSPS KAKENHI, Grant Number JP19H04082.

**Data Availability Statement:** Case studies and source code in this study are publicly available at https://github.com/yatiphyo/DCA2MC, accessed on 29 May 2023.

**Conflicts of Interest:** The authors declare no conflicts of interest.

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
