# Peer review of "A Layered and Parallelized Method of Eventual Model Checking"

_information, doi:10.3390/info14070384_

Round 1

Reviewer 1 Report

*Summary*

The authors present an approach to the parallelization of model checking eventual LTL properties, e.g. termination.
They divide the model checking problem into independent problems using a divide and conquer approach that they call DCA2EMC. The construction of these sub-problems is performed in a layered way according to the reachability from the initial states.
They present a tool based on Maude model checker. The tool and experimental results are available on github. The experiments have been performed on 5 classical protocol benchmarks. They compare DCA2EMC in its sequential and parallel versions with baseline Maude. They also evaluate the scaling with the number of parallel workers.

*Evaluation*

Pros
- Relevant topic
- Quite well-written paper
- Appears to be technically sound

Cons
- Some concepts should be explained in more detail.
- The paper is based on many previous publications by the authors, in particular how the layer specifications are determined. It would be easier to understand the algorithm if the paper was more self-contained. A concrete example of concepts such as "sub-space" should be provided.
- It is not explained what this article adds upon the conference publication [11].

*Comments and Questions*

page 1: I don't quite understand what is the difference between the divide and conquer approach presented in this article vs the divide and conquer approaches for other properties. Which specific aspects of the properties do you exploit? In particular what is the difference to [9] and [10]?

page 2: "benefit from multi-core architecture": Model checking is usually memory-bound. Did you observe that memory access is a limiting factor on parallelization?

page 2: What are the extensions of this article over [11]?

page 3/paragraph before section 3: It would be great to give concrete examples for the "soup" definitions as this is a non-standard/unfamiliar notation and terminology.

page 4: What is a "final layer"?

page 4: What's the difference between "layered" and "breadth-first"?

page 4: If I understand correctly the "layering" is just performed to determine "initial states" of the Lth layer. The model checking problems from each of these Lth layer "initial states" can then be parallelized.
(1) I find it confusing to talk about a "layered approach" because the layering has nothing to do with the parallelization itself. The parallelization is not in layers but rather in "slices".
(2) What is a "sub space" then? Is it just a single Lth layer "initial state"? Can it contain several states?
(3) Could these Lth layer "initial states" just be explained as "states reachable from an initial states within L steps? That would get rid of the confusing "layering" explanation.

page 8: What determines the layer specification (e.g. 3 3 3)? What is the effect of choosing different specifications? This seems crucial for the technique. Can you elaborate on this?

page 9: Would partial order reduction avoid the blow up in the TAS benchmark?

page 1: broken reference for stable properties

page 2: PSIN -> SPIN

page 2: making us easier -> making it easier???

page 3/section 3: in order to model checking -> in order to model check

page 4: L and then n_l -> L then n_l

page 5: broken reference for parallelization overhead

page 10: SAT/SM -> SAT/SMT

Author Response

Please check our response to the reviewer in the attached file.

Reviewer 2 Report

This paper proposes a divide-and-conquer approach to parallelly split an original model-checking problem into multiple smaller model-checking problems based on a master-worker pattern and implemented in the Maude framework. This paper extends a DSA conference 2022 paper on a tool for model-checking eventual properties called DCA2EMC. The authors implement a parallel version of DCA2CSMC. The experimental results show the effectiveness of the approach and the comparison among the sequential tool, the parallel tool, and the Maude LTL Model model checker.

----------- MAIN STRENTHS AND WEAKNESSES -----------

+ The paper presents a parallel version of a divide-and-conquer approach for eventual model checking with a master-worker pattern.

+ The parallel version is interesting and potentially useful.

- There are several presentation issues, including reference ones.

- There are some writing problems.

----------- COMMENTS -----------

The paper proposes a layered and parallel technique/tool based on a master-worker pattern for the divide-and-conquer approach to eventual model checking. The results demonstrate the running performance improvement of the tool compared with the sequential tool for five protocols and significantly surpass Maude LTL model checker for four protocols. However, I have to make some comments.

First, the presentation of the paper can be improved. (1) In the third paragraph of Section 4, "In the code fragment at lines 26–28" should be "In the code fragment at lines 17–20". The reason is that there are no lines 26–28 in Algorithm 2. (2) In Table 1, it is better that columns are "Protocol", "Sequential tool (DCA2EMC)", "Parallel DCA2EMC (8 workers)", "Maude LTL model checker", and "Layers" according to the table title. (3) Table 2 can only include two columns: "#Worker" and "Time Cost" or "Running Performance". The author can make Table 2 more concise and more consistent with Figure 3 by describing the information about protocol, layers, and the performance of Maude LTL model checker and DCA2EMC in the paragraph of Section 5.2 rather than in Table 2. 

Furthermore, there are some reference problems in the paper. (1) In the second paragraph of Section 1, there is an invalid reference: "and until and until stable properties [?]". (2) In the first paragraph of Section 4, there is an invalid reference: "from parallelization as demonstrated in [?]". (3) In references, most references have URLs, such as https://doi.org/10.1007/978-3-319-10575-8. However, some references (e.g., 10, 13, 19, and 23) do not have URLs. All references should be consistent.

English:

There are some writing issues that should be fixed in the paper. 

(1) Section 1: 2nd paragraph: "and until and until stable properties" should be "and until stable properties".

(2) Section 1: 2nd paragraph: "in linear temporal logic (LTL), however, we need " should be "in linear temporal logic (LTL); however, we need ".

(3) Section 1: 4th paragraph: "[14] and so is the parallel tool" should be "[14], as has the parallel tool".

(4) Section 1: 4th paragraph: "PSIN are comparable [18]" should be "SPIN are comparable [18]".

(5) Section 4: 1st paragraph: "each non-final layer of DCA2EMC can essentially be conducted independently, however," should be "each non-final layer of DCA2EMC can essentially be conducted independently; however,".

(6) Section 4: 1st paragraph: "can totally be conducted" should be "can be totally conducted".

(7) Section 5: 1st paragraph: "Because Maude LTL model checker and SPIN" should be "Maude LTL model checker and SPIN". "Because" should be removed due to a subsequent word "Hence".

(8) Section 7: 1st paragraph: "by Maude sockets" should be "through Maude sockets".

Author Response

please check our response to the reviewer in the attached file

Round 2

Reviewer 1 Report

I have reviewed a previous version of the paper.

The authors have clarified or addressed most issues raised.

However, I reiterate my comment from the first review that making the article more self-contained would increase its value to the reader:

The paper is based on many previous publications by the authors, in particular how the layer specifications are determined. It would be easier to understand the algorithm if the paper was more self-contained.